# Research on parameter identification of shaking table systems based on the RLS method

Chunhua Gao, Yanping Yang*, Mengyuan Qin, Cun Li, Zihan Yuan

College of Architecture and Civil Engineering, Xinyang Normal University, Xinyang, Henan, China

* hhy0411ping1013@163.com

## Abstract

It is difficult to accurately establish a model of the real mesa system. Furthermore, a model of a seismic simulation vibration table array system is critical to increasing the accuracy of seismic testing in laboratory settings. Herein a model of the nine subarray shaking table system is identified by recursive extension of the least square method, which is used to accurately identify the structure parameters by simulation of the structure assuming a single degree-of-freedom. Then, through the displacement of the empty shaking table and the application of the recursive least squares algorithm, the model of the seismic simulation vibration table array is established. Through this study, the vibration table model of different construction forms can be obtained, and the parameters that are difficult to measure for some complex structures can effectively be determined.

**Data Availability Statement:** All relevant data are within the paper and its Supporting Information files.

**Funding:** GCH was supported by the Key Science and Technology Project of the Science and

## 1. Introduction

The shaking table is a very important device for simulating earthquakes in the laboratory and provides an accurate method for measuring structural responses under seismic conditions [1–4]. With the development of the shaking table, it has become an effective measure for studying the elastoplastic seismic response of structures [5]. According to different driving sources, it can be divided into electromagnetic drive and actuator drive, among which the actuator drive vibration table includes electric vibration table and electro-hydraulic servo vibration table [6]. Electro-hydraulic servo shaking table is widely used in various fields, such as civil, aerospace and nuclear engineering, because of its advantages of large output, large stroke and fast response. Ahmad et al. [7] established a scale model of a two-story reinforced concrete frame structure with a 1/3 scale coefficient, and conducted a full dynamic shaking table test on it. The study found that when low-strength concrete was used and no node limit band was set, the safety of the joint was difficult to be guaranteed. It is verified that the ASCE ACI 352 node shear strength model overestimates the shear strength of low strength concrete.

However, in the modelling process of the electrohydraulic servo system of the hydraulic vibration table, there are many nonlinear factors that cannot be well reflected in the model, such as the dead zone of the servo valve, friction between components, and clearance, which

Technology Department of Henan Province (182102210539), the Key Research Project of Henan Province (No 18B560009), (No 23A560015) and the Nanhu Young Scholars Program of Southwest Normal University, while YYP was supported by the Graduate Innovation Fund Project of Xinyang Normal University (No 2021KYJJ75). The funders had no role in study design, data collection and analysis, decision to publish, or preparation of the manuscript.

**Competing interests:** The authors have declared that no competing interests exist.

cause the system model to exhibit a considerable degree of uncertainty [8]. In addition, the dynamic characteristics of the servo system are also affected by factors such as equipment manufacturing, oil supply pressure, and level of input signal, so that the output waveform of the exciter is different from the expected waveform, and the waveform reproduction ability is weak [9, 10]. To solve these problems, many scholars have proposed an identification method to determine the parameters that are difficult to calculate.

Many identification methods are commonly used in engineering, such as neural network, support vector machine, Bayesian estimation and least square method [11–13]. Among them, the neural network algorithm adjusts the similarity between the output and the expectation by adjusting the connection weight between neurons, which requires a large number of data sets, and its own interpretability is relatively poor, and it is easy to fall into the local optimal solution [14, 15]. Based on statistical theory and structural risk minimization principle, support vector machine algorithm has global optimality and good "robustness". However, it is difficult to implement large-scale training samples, and is sensitive to the selection of parameters and kernel functions, with great uncertainty [12, 16]. Bayesian estimation uses Bayesian theorem combined with prior information and new observation information to estimate the new probability of occurrence, which to some extent solves the problem that classical estimation cannot be applied to solving the probability of non-repeatable independent events. However, it relies on the distribution state of prior information and has poor objectivity [17]. The least squares method is widely used in the field of system model parameter identification by minimizing the sum of squared errors to obtain the best matching function, which is simple, good analytical and unique in optimal solution [13, 18, 19].

Compared with the least square method, the recursive least square method (RLS) can be used for real-time identification with less computational effort Luo Qinqin et al. [20] identified the main parameters of the virtual synchronous generator by using recursive least square method, and the results show that the identified parameters meet the accuracy requirements. Lin Juguang et al. [21] proposed an improved recursive least square method for online identification of the inductance parameters of the D-axis and the Q-axis of permanent magnet synchronous motor, and achieved efficient identification results. In this paper, nine sub-table system is taken as the research object, and RLS algorithm is used to identify inherent parameters of the shaking table that will change during the experiment, such as the mass and stiffness. Through different test data, the parameters of shaking table are estimated by RLS algorithm, and the response of shaking table is rebuilt. The surface response of shaking table reconstructed by RLS method is compared with the surface response of shaking table measured by test, and the final model of shaking table is determined. It is found that the identification of the parameters of the shaking table by the RLS method is accurate and available and can provide an important reference for future shaking table research.

## 2. System description

This paper takes the nine-subarray shaking table system of the Beijing University of Technology as the research object, as shown in Fig 1. The table support system is shown in Fig 2. The modular array system can be placed freely to form a large vibration table for testing or can be used for multiple shaking tables for array testing. Each sub-shaker consists of four components, namely the table, shaker, connecting rod and base, and each sub-shaker can be mounted vertically or horizontally at any time to achieve different seismic wave input and motion modes. For example, a shaker is installed in the x direction, two connecting rods in the y direction, and three connecting rods in the z direction to achieve unidirectional motion, as shown in Fig 3(A); Install one shaker and two shakers in the x direction and y direction respectively,

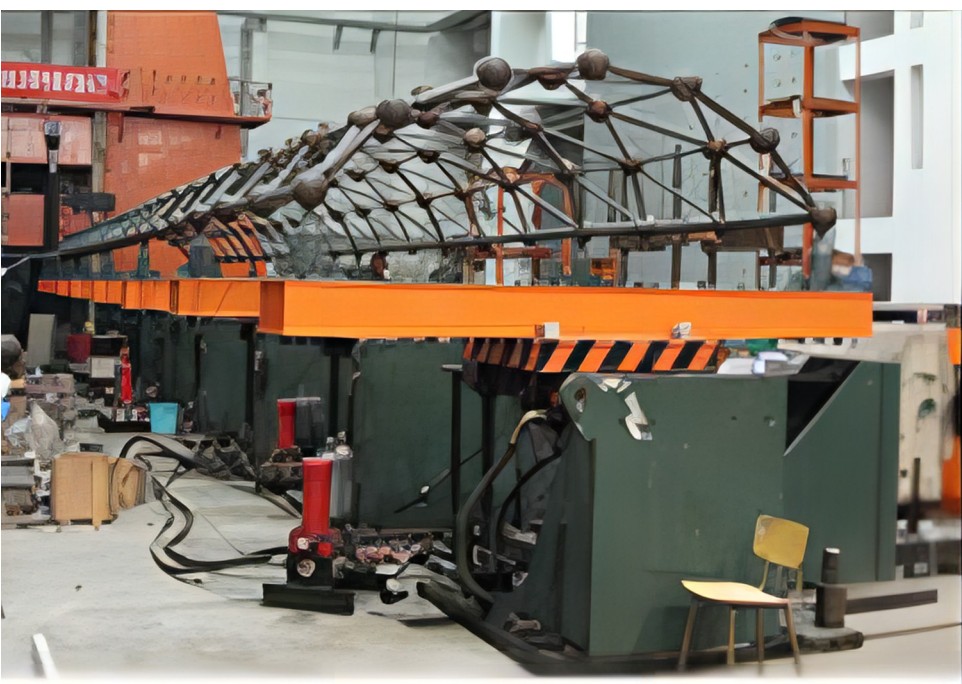

**Fig 1. Array system.**

and install three connecting rods in the z direction to realize the three degrees of freedom movement of double horizontal plus rotation, as shown in Fig 3(B); By installing one shaker in the x direction, two shakers in the y direction and three shakers in the z direction, three-direction and six-DOF motion can be realized, as shown in Fig 3(C) [22]. The layout of array system is flexible, and the typical layout of the array is shown in Fig 4(A)–4(D). Table 1 shows the skill performance indices of the nine subarray systems. Fig 5 is a conceptual diagram of the entire control system, which is a host-target style control system. The host computer serves as a human-computer interaction interface and allows users to enter command signals and displays the system's current state and data.

## 3. Recursive least squares algorithm

Widrow and Hoff [23] developed the LSM algorithm in 1960; later, many scholars improved it, forming the recursive least square method (RLSM). Because of its computational simplicity, it has been widely applied in many fields, such as control systems, system identification, and data analysis. The recursive least square method algorithm can be written as Eq (1):

$$y(k) = \theta\varphi^T(k) + e(k) \tag{1}$$

where $\theta$ is the parameter to be estimated, $e(k)$ is the prediction error signal, and $\varphi(k)$ is the vector data. In the procedure of system identification, $y(k)$ and $\varphi(k)$ are the measured experimental data, and the prediction error is minimized. The performance index is defined in Eq (2):

$$J = \sum_{k=1}^{L} \lambda^{L-k}[y(k) - \varphi^T(k)\hat{\theta}]^2 \tag{2}$$

where $\lambda$ is the forgetting factor ($0 < \lambda \leq 1$).

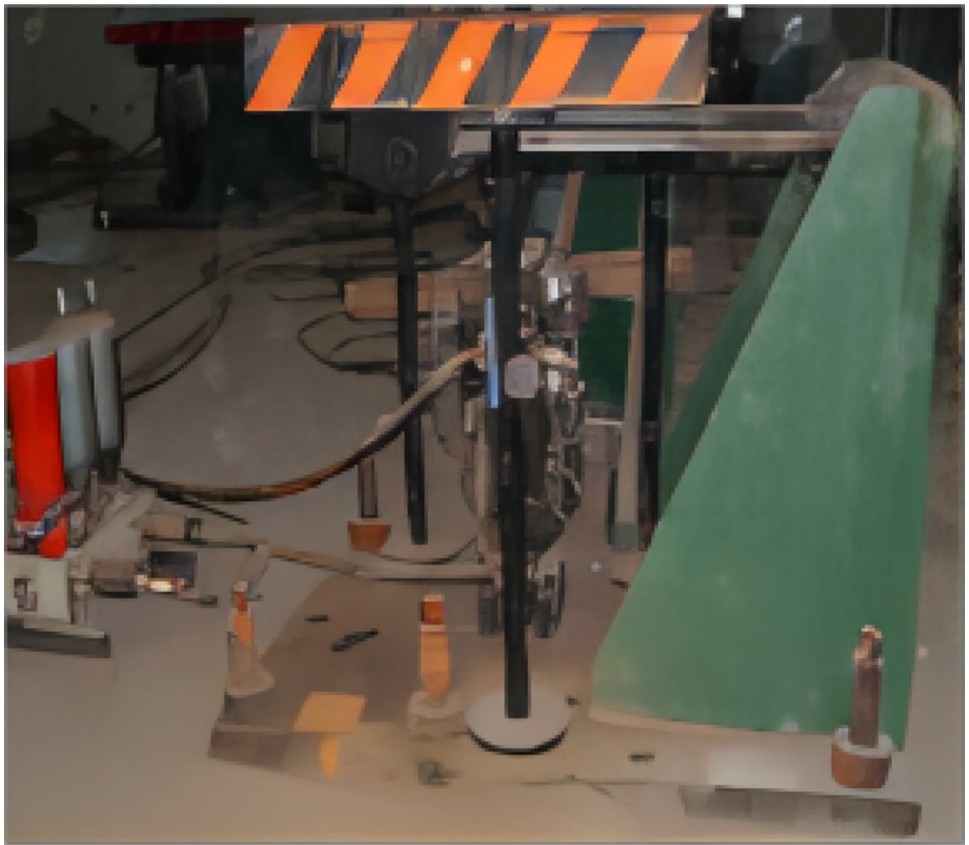

**Fig 2. Mesa support system.**

In line with the matrix principle, Eq (2) can be expressed as Eq (3):

$$J = eWe^T \tag{3}$$

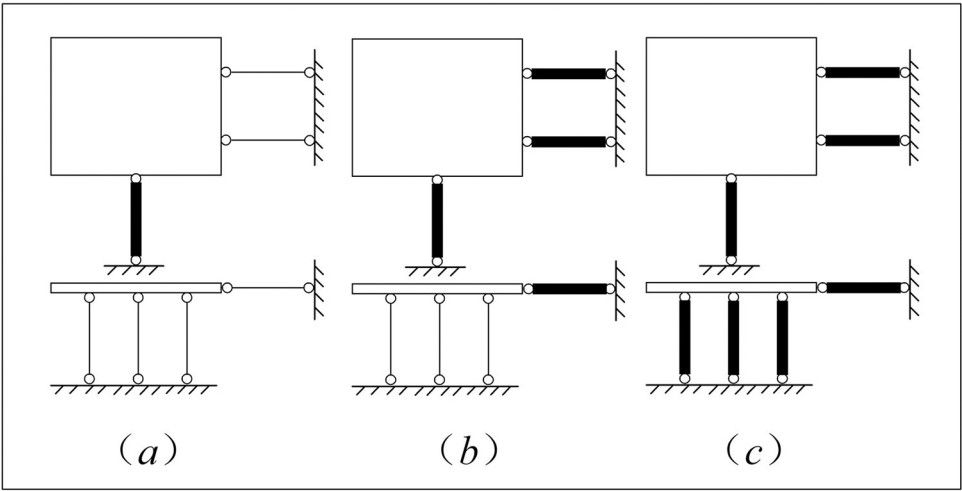

**Fig 3. Single shaking table movement mode.**

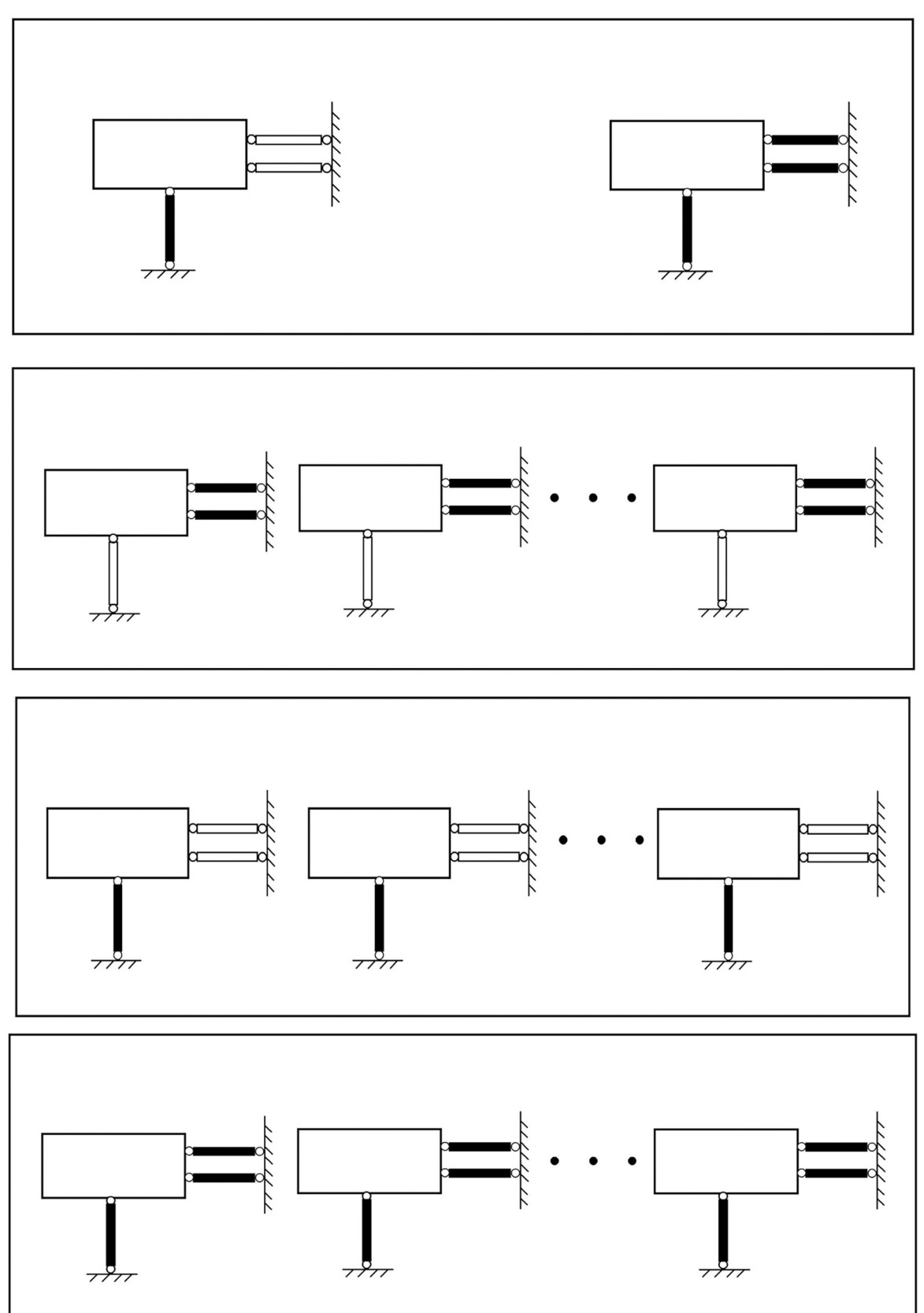

**Fig 4. Typical arrangement of the array table.** (a) Dual array system; (b) Y—moving multi-array system; (c) x—moving multi—array system; (d) Dual horizontal motion multi array system.

where $e = [e(1) \; e(2) \cdots e(k)]$, $W$ is the data weight matrix, $e(k) = y(k) - \hat{y}(k-1)$, and $\hat{y}(k-1) = \theta \varphi^T(k)$. Thus, parameter estimation of the recursive least squares algorithm is obtained, as shown in Eq (4):

$$\begin{cases} \hat{\theta}_k = \hat{\theta}_{k-1} + K(k)[y(k) - \varphi^T(k)\hat{\theta}_{k-1}] \\ K(k) = \dfrac{P(k-1)\varphi(k)}{\lambda + \varphi^T(k)P(k-1)\varphi(k)} \\ P(k) = \dfrac{1}{\lambda}[I - K(k)\varphi^T(k)]P(k-1) \end{cases} \quad (4)$$

where $K(k)$ is the correction factor, $P(k)$ is the covariance matrix, and $\lambda$ is the forgetting factor (where the forgetting factor must choose a positive number close to 1).

Step size signifies a compromise between the rate of convergence and final maladjustment [24]. In practical applications, it is usually very difficult to select the step size. From the above derivation, a schematic diagram of recursive least square system identification is shown in Fig 6.

From Fig 6, it can be seen that the system parameters $\theta(k)$ and the covariance matrix $P(k)$ of recursive least squares system identification will be updated in every operation; therefore, in the initial operation, the system parameters and the covariance matrix are given the initial values, defined as $P(0) = \alpha I$, where $\alpha$ is a sufficiently large positive real number, generally $10^4 \sim 10^{10}$, and $\varepsilon$ is the zero vector or sufficiently small positive real vector.

## 4. Recursive least square method for parameter identification of the array system

The mechanical model of the shaking table can be represented by Eq (5):

$$M_e \ddot{u}_x(t) + K_e u_x(t) + (C_e|\dot{u}_x(t)|^{\alpha} + F_{\mu e})sign(\dot{u}_x(t)) = F_{act}(t) \quad (5)$$

where $F_{act}(t)$ is the horizontal actuator propulsion and $M_e$ is the effective mass matrix. $M_e =$

$\begin{bmatrix} M_1 & 0 & \cdots 0 \\ 0 & M_2 & \cdots 0 \\ 0 & 0 & \cdots M_3 \end{bmatrix}$ (not considering the coupling between them), $K_e$ is the horizontal effective stiff-

ness matrix $K_e = \begin{bmatrix} K_1 & 0 & \cdots 0 \\ 0 & K_2 & \cdots 0 \\ 0 & 0 & \cdots K_3 \end{bmatrix}$, $C_e$ is the effective damping matrix $C_e = \begin{bmatrix} C_{11} & C_{12} & \cdots C_{13} \\ C_{21} & C_{22} & \cdots C_{23} \\ C_{31} & C_{32} & \cdots C_{33} \end{bmatrix}$,

**Table 1. Performance indices of the 9-table seismic array.**

| Table Size | Table weight | Maximum load | Frequency | Max velocity |
|---|---|---|---|---|
| 1 m×1 m | 1Ton | 5Ton | 0.4~50 Hz | 60 cm/s |
| **Displacement accuracy** | **Actuator stroke** | **Actuators number** | **Control** | **Max acceleration** |
| ≤0.005 | ±75mm | 16 | acceleration | 2.0 g/1.0 g |

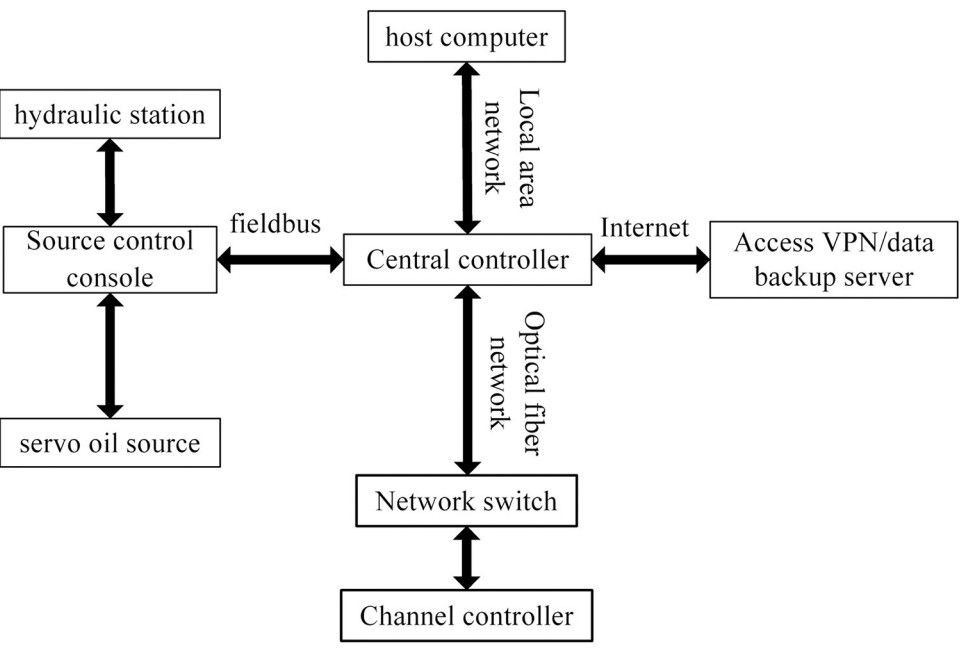

**Fig 5. The control system.**

$F_{\mu e}$ is the frictional force, $u_x(t)$ is the longitudinal horizontal displacement, and $\alpha$ is an undetermined constant.

Its objective function should be set as the smallest normalized error, as shown in Eq (6):

$$\frac{\sum_{i=1}^{N} \lambda_i \int_0^{T_i} \left[ F_{act}(t) - M_e\ddot{u}_x - K_e u_x - (C_e |\dot{u}_x|^\alpha + F_{\mu e}) sign(\dot{u}_x) \right]^2 dt}{\sum_{j=1}^{N} \lambda_j \int_0^{T_j} F_{act}^2(t) dt} \tag{6}$$

where $\nabla$ is the normalized error, $N$ is the number of tests, $T_i$ is the duration of the first test, $\lambda_i$ is the weight distribution of the first test, and the remaining symbols are defined as previously stated above.

The standardization of Eq (6) can be represented by Eq (7):

$$\nabla^2 = 1 + y^T a y - b^T y - y^T b \tag{7}$$

where $y^T = (M_e, K_e, C_e, F_{\mu e}), a = \sum_{i-1}^{N} \bar{\lambda}_i \int_0^{T_i} RR^T dt, \ b = \sum_{i-1}^{N} \bar{\lambda}_i \int_0^{T_i} RF_{act}(t)dt$, among which

$R^T = (\ddot{u}_x(t), u_x(t), |\dot{u}_x(t)|^\alpha sign(\dot{u}_x(t)), sign(\dot{u}_x(t)))$, $y$ is the identified parameter vector and $R$ is the response vector, which can be obtained from the measured experimental data.

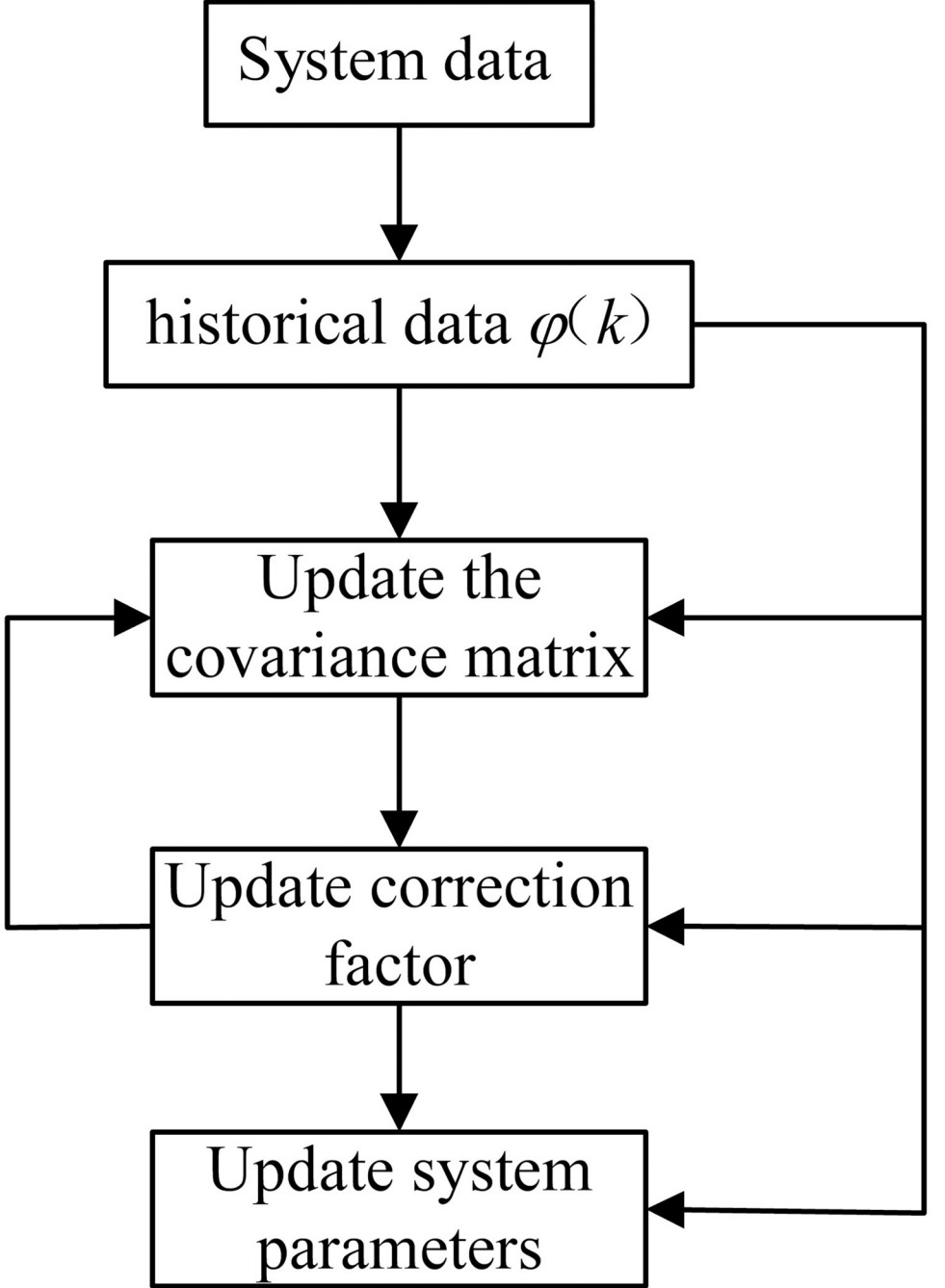

**Fig 6. Schematic diagram identification system using the least squares method.**

The normalized weight coefficient $\bar{\lambda}_i$ can be defined by Eq (8):

$$\bar{\lambda}_i = \frac{\lambda_i}{\sum\limits_{j=1}^{N} \lambda_j \int\limits_0^{T_j} F_{act}^2(t)dt} \tag{8}$$

In this paper, tests of different amplitudes are carried out on the array system, so the weight coefficient of each test should be considered when calculating the normalized error. The standard error of the ith test is set as Eq (9):

$$\frac{\int_0^{T_i} [F_{act}(t) - M_e\ddot{u}_x - K_e u_x - (C_e|\dot{u}_x|^\alpha + F_{\mu e})sign(\dot{u}_x)]F_{act}(t)dt}{\int_0^{T_i} F_{act}^2(t)dt} \tag{9}$$

The square of the minimum normalized error can be shown as Eq (10):

$$\overline{\nabla}^2 = \sum_{i=1}^{N}[\bar{\lambda}_i\left\{\int_0^{T_i} F_{act}^2(t)dt\right\}\nabla_i] \tag{10}$$

In general, there are two forms for the evaluation of the test weight coefficient $\lambda_i$, one of which can be expressed as Eq (11) and the other as Eq (12):

$$\lambda_i = 1/\int_0^{T_i} F_{act}^2(t)dt \tag{11}$$

$$\lambda_i = 1 \tag{12}$$

When $\lambda_i = 1\left/\int_0^{T_i} F_{act}^2(t)dt\right.$, Eq (10) can be adapted as Eq (13):

$$\overline{\nabla}^2 = \frac{1}{N}\sum_{i=1}^{N} \nabla_i \tag{13}$$

As we can see from Eq (12), if the weight coefficient $\lambda_i = 1/\int_0^{T_i} F_{act}^2(t)dt$, the importance of each test has no relation to the horizontal actuator propulsion in the test.

In another case, when the weight coefficient $\lambda_i = 1$, Eq (10) can be expressed as Eq (14):

$$\overline{\nabla}^2 = \sum_{i=1}^{N}(\int_0^{T_i} F_{act}^2(t)/\sum_{j=1}^{T_j} F_{act}^2(t)dt)\nabla_i \tag{14}$$

It can be seen from Eq (14) that the square of the normalized error is related to the output of the exciter in each experiment; the greater the output of the exciter is, the greater the weight, or the smaller the weight.

## 4.1 Undetermined parameter $\alpha$

In this paper, six groups of sinusoidal waves with different frequencies (Table 2) are used to test the array system, and the RLS method is used to identify the parameter $\alpha$. To optimize the parameter $\alpha$, a series of error mean values can be obtained from different values of $\alpha$. In this

**Table 2. Test system feature on sine waves.**

| condition | Fre (Hz) | Dis (cm) | Vec (cm/s) | Acc(cm/s$^2$) |
|-----------|----------|----------|------------|----------------|
| 1 | 0.04 | 4 | 1 | 0.294 |
| 2 | 0.05 | 4 | 1.51 | 0.588 |
| 3 | 0.10 | 4 | 2.51 | 1.568 |
| 4 | 0.4 | 4 | 10.05 | 63.014 |
| 5 | 0.8 | 10 | 50.24 | 252.154 |
| 6 | 1.0 | 4 | 25.12 | 157.587 |

paper, values between 0 and 1 are taken as $\alpha$, and the oil source pressure in the test is $15MPa$. The weight coefficients are defined as follows:

$$\lambda_i(1) = 1, \lambda_i(2) = 1/\int_0^{T_i} (u_x)^2 dt \ \lambda_i(3) = 1/[\int_0^{T_i} (u_x)^2 dt]^{1/2}, \lambda_i(4) = 1/\int_0^{T_i} F_{act}^2(t)dt.$$ The

curve of normalized error varying with $\lambda_i$ and $\alpha$ is shown in Fig 7.

From Fig 7, it can be concluded that the normalized error mean changes with the change in $\alpha$ and weight $\lambda_i$, but it is not particularly obvious. In this paper, the changes in Coulomb friction in each working condition are shown in Fig 8(A) and 8(B) and viscous damping coefficient in each working condition are shown in Fig 9(A) and 9(B), respectively, under the two conditions of $\alpha$ = 0.5 and $\alpha$ = 1.0.

## 4.2 Effective mass

The $\alpha$ and test weight coefficients are the same as above. White noise, sinusoidal waves and El Centro (NS) seismic waves are used for the experiment. Table 3 represents the effective mass of the oil source pressure identification and estimation system. The effective mass of the system is estimated by RLS method under oil source pressures of $10MPa$ and $15MPa$.

As shown in Table 3, for distinct test waveforms, oil source pressure and weight coefficient, the estimated values of effective quality are almost the same. It can also be seen that with the same test waveform, the effective mass of the system increases with increasing oil source pressure because there is a certain correlation between the equivalent mass of the system and the equivalent stiffness. Under the same oil source pressure, the average effective mass of the system in the white noise test with different energy levels is 1.137 t, while the average effective

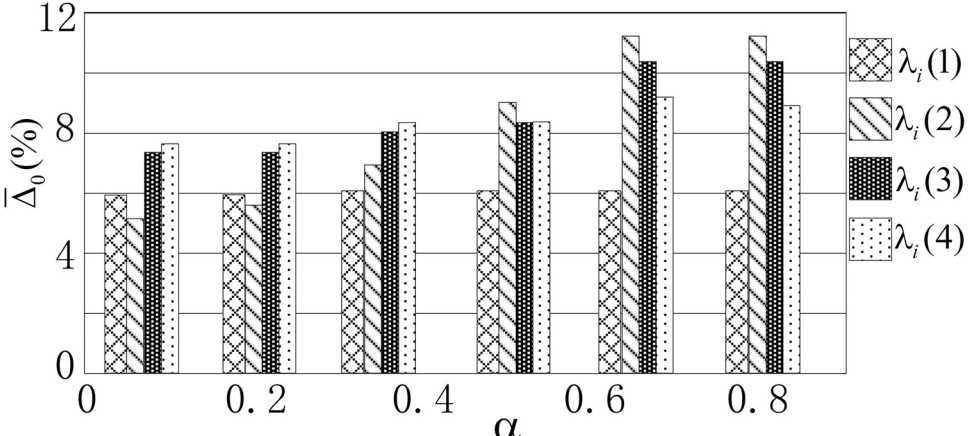

**Fig 7. Curve relationship with normalized average error and different parameters.**

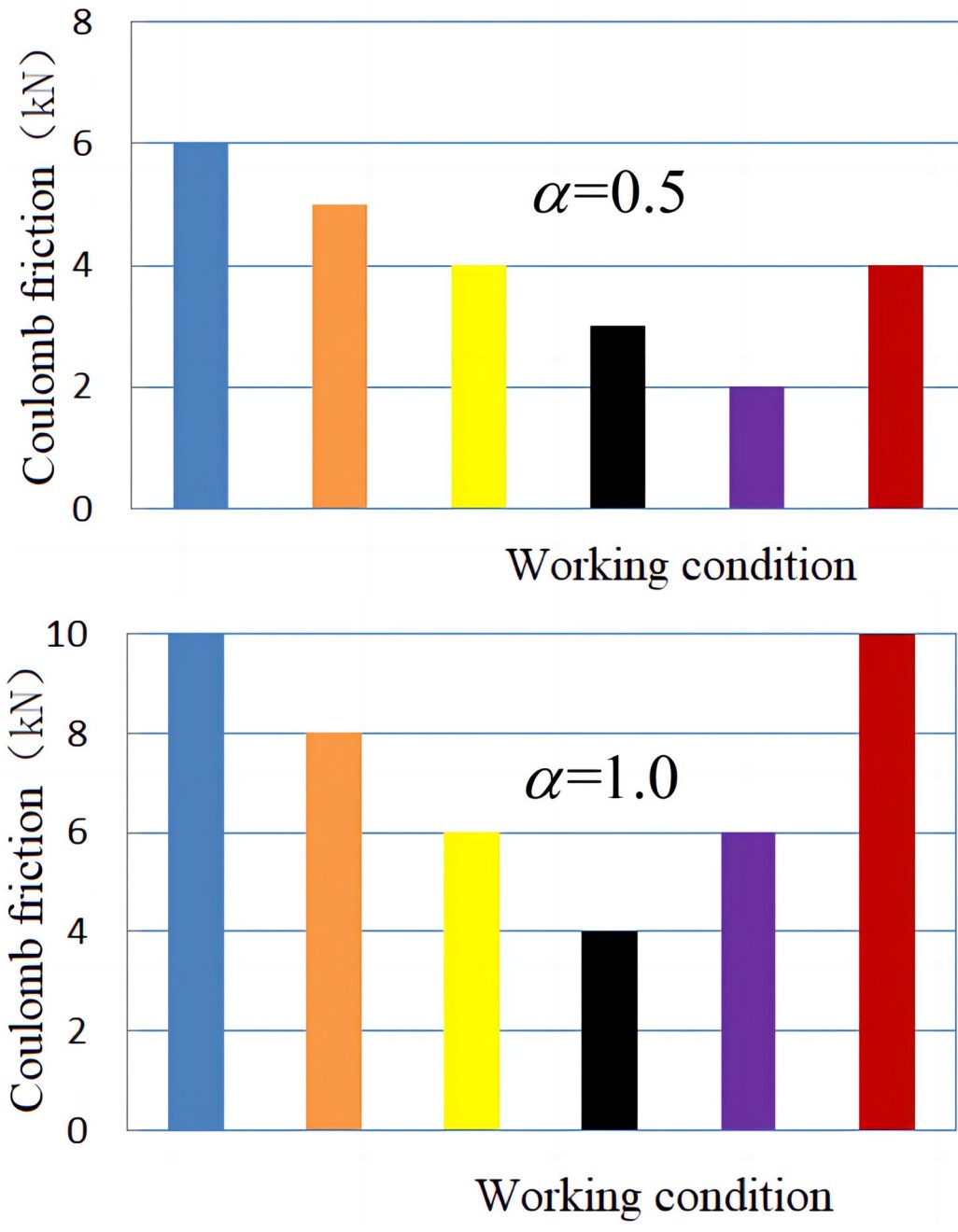

**Fig 8. Coulomb friction curve under different working conditions.** (a) $\alpha = 0.5$; (b) $\alpha = 1.0$.

mass of the system in the sinusoidal wave and El Centro seismic wave tests is 1.145 t, which is generally coincident with the white noise test conducted at different energy levels.

### 4.3 Effective stiffness

White noise, sinusoidal and electric-centro seismic waves with small energy levels (0.03 g) and large energy levels (0.10 g) are used to estimate the horizontal effective rigidity of the array system. When the experimental weight coefficients are $\lambda_i = \lambda_i(1), \lambda_i = \lambda_i(2)$ and $\alpha = 0.5$, the

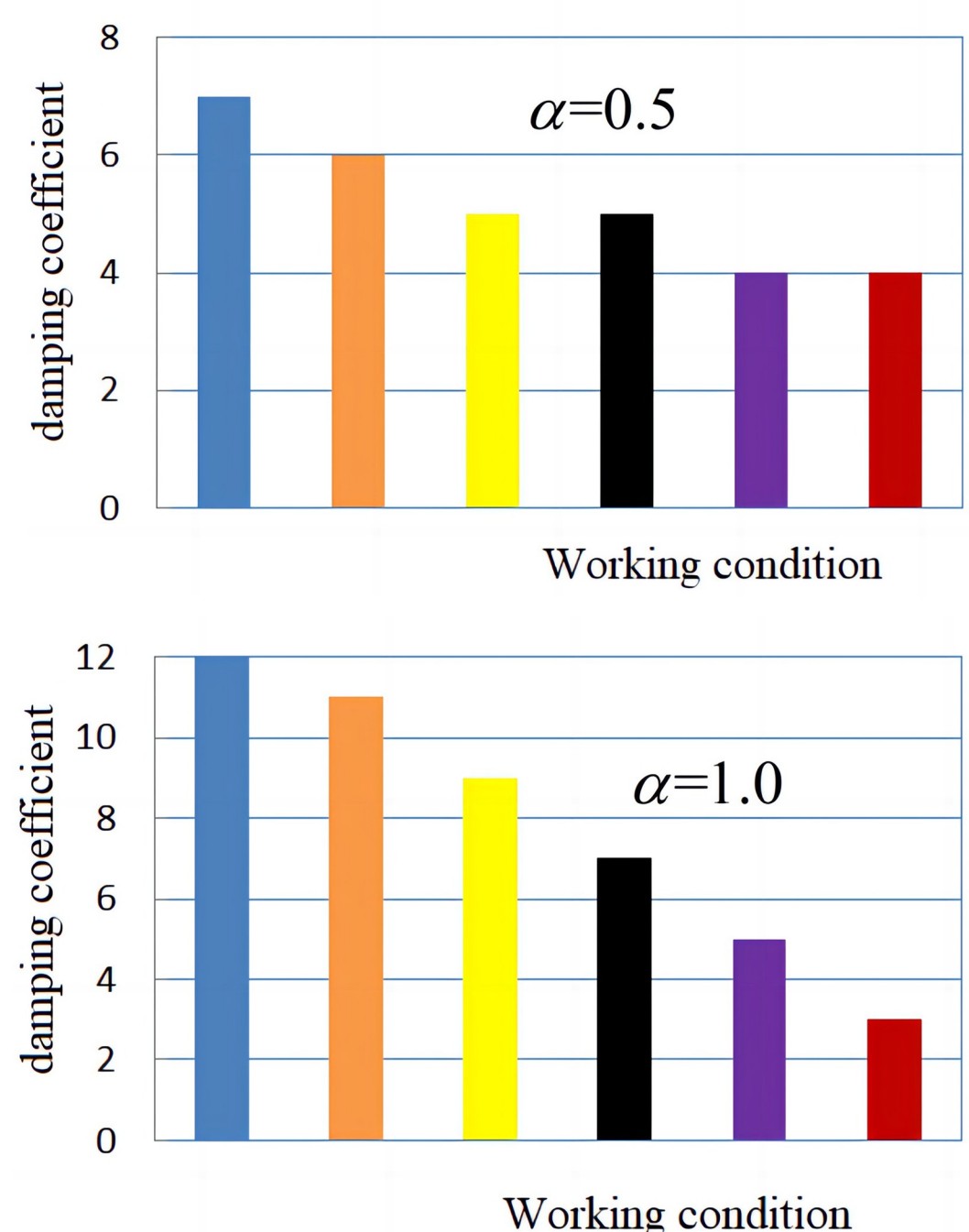

**Fig 9. System damping coefficient curve under different conditions.** (a) $\alpha = 0.5$; (b) $\alpha = 1.0$.

horizontal effective stiffness of the system is estimated, and Table 4 indicates the estimated horizontal effective stiffness of the system.

Table 4 shows that the effective horizontal stiffness of the white noise system with different energy levels is 9.232 kN/m. With increasing oil source pressure, the effective stiffness of the system increases, while the measured effective stiffness of the system by the seismic wave is slightly lower than that by the sinusoidal wave test.

Table 3. Effective quality system ($\alpha$ = 0.5).

| test waveform | oil source pressure (MPa) | Effective mass (Ton) ($\lambda_i = \lambda_i(1)$) | Effective mass (Ton) ($\lambda_i = \lambda_i(2)$) |
|---|---|---|---|
| White noise (0.03 g) | 10 | 1.136 | 1.135 |
| White noise (0.10 g) | 10 | 1.142 | 1.142 |
| sinusoidal | 10 | 1.132 | 1.135 |
| sinusoidal | 15 | 1.139 | 1.145 |
| el-centro (NS) | 15 | 1.162 | 1.157 |

## 4.4 Shaking table parameter identification by the recursive least squares algorithm

To simplify the identification of the shaking table parameters, it is assumed that the vibration platform of the shaking table array system induces only a one-way motion, which can be simplified into a plane structure of degrees of freedom, and the equation of motion can be defined by Eq (15):

$$M(t)\ddot{x}(t) + C(t)\dot{x}(t) + K(t)x(t) = F(t) \tag{15}$$

where $M(t)$ is the mass matrix of the shaking table array system, $C(t)$ is the damping matrix, $K(t)$ is the stiffness matrix, $x(t)$ is the displacement of the table, and $F(t)$ is the applied load of the table. By measuring the displacement of the table, the $M$, $C$, $K$ in the earthquake shaking table array system is identified, and then the model is modified.

The state vector is defined in Eq (16):

$$Z(t) = [x(t), \dot{x}(t), M(t), C(t), K(t)]^T \tag{16}$$

The equation of state of the shaking table array system is expressed as Eq (17):

$$\frac{dZ(t)}{dt} = f[Z(t), t] + \xi(t) = [\dot{x}(t), \ddot{x}(t), \dot{M}(t), \dot{C}(t), \dot{K}(t)]^T + \xi(t) \tag{17}$$

It is assumed that the displacement of the shaking table array system is standard, and the observation equation of the array system can be shown by Eq (18):

$$Y(i+1) = [1, 0, 0, 0, 0, 0] \times Z(i+1) \tag{18}$$

Table 4. System effective stiffness ($\alpha$ = 0.5).

| test waveform | oil source pressure (MPa) | Effective stiffness (kN/m) ($\lambda_i = \lambda_i(1)$) | Effective stiffness (kN/m) ($\lambda_i = \lambda_i(2)$) |
|---|---|---|---|
| White noise (0.03 g) | 10 | 9.625 | 9.843 |
| White noise (0.10 g) | 10 | 8.52 | 8.940 |
| sinusoidal | 10 | 6.73 | 6.802 |
| sinusoidal | 15 | 9.46 | 9.511 |
| el-centro (NS) | 15 | 9.454 | 9.4601 |

According to the observation equation of the array system, the optimal state quantity of the array system is calculated by the recursive least square algorithm, thus the model of the array system is modified.

## 4.5 Simulation example

In an N degrees of freedom structure simulation model, the recognized system parameters are mass, damping and stiffness, and ground motion El Centro (NS) waveforms are used with a peak acceleration of 0.33$g$ during simulation. This paper adopts the recursive least squares identification method to identify and estimate the array system.

In this paper, the model is established and simulated. In the simulation, El Centro (NS) is used as the input waveform and recursive least squares identification algorithm. The displacement output is taken as the observed value to verify the correctness of the identification algorithm. Fig 10(A)–10(C) is the simulation result.

It can be seen from Fig 10(A)–10(C), the recursive least squares identification algorithm is reasonable for identifying the stiffness, damping and mass of the system.

To further check the correctness of the algorithm, the real value of shaking table displacement obtained by the experiment and the identification value of the recursive least squares algorithm are shown in Fig 11.

As we can see from Fig 11, the true value of the shaking table displacement agrees well with the identification value obtained by the recursive least squares identification algorithm, which further confirms the exactness and accuracy of the recursive least square identification method proposed in this paper. Fig 12 shows the simulation results.

Fig 12 shows that the measured values of the shaking table acceleration are basically consistent with the calculated values of the system parameters obtained by the recursive least squares identification method, and the deviation between the surveyed values and count values is approximately 8.96%, which is considered relatively accurate.

## 4.6 Experimental validation

Based on the experimental study of the nine-subarray system of the Beijing University of Technology earthquake simulation platform, the mechanical model is simplified to a 9-DOF structure, and the mass, damping and stiffness of the nine-subarray system are identified by the recursive least squares identification algorithm. An El Centro (NS) waveform with different displacement peaks was used as the input wave in the test, and Table 5 demonstrates the test conditions. In the experiment, acceleration sensors were arranged on the table to measure the table acceleration. Fig 13(A) and 13(B) shows the measured and calculated values of acceleration and the corresponding error values. The test identification results of three groups of different displacement peaks are shown in Table 6.

Fig 13(A) and 13(B) shows that the measured value of acceleration agrees well with the calculated value, which confirms the correctness of parameter identification.

It can be seen from the identification results of the array system in Table 6 that the identification results of the three groups of tests with different displacement amplitudes are very close, and it can be inferred that the mass is 1.137$t$, the stiffness is 9.427$kN/m$ and the damping of the seismic simulated array system is approximately 0.941$kN/m/s$.

The acceleration time-history curve is shown in Fig 14(A) and 14(B):

As seen in Fig 14(A) and 14(B), the array system acceleration error between the surveyed values and count values is approximately 16.8%, the array system displacement error between calculated values and measured values of displacement error is approximately 10%, and the acceleration time history curve has good alignment. From these results it can be concluded

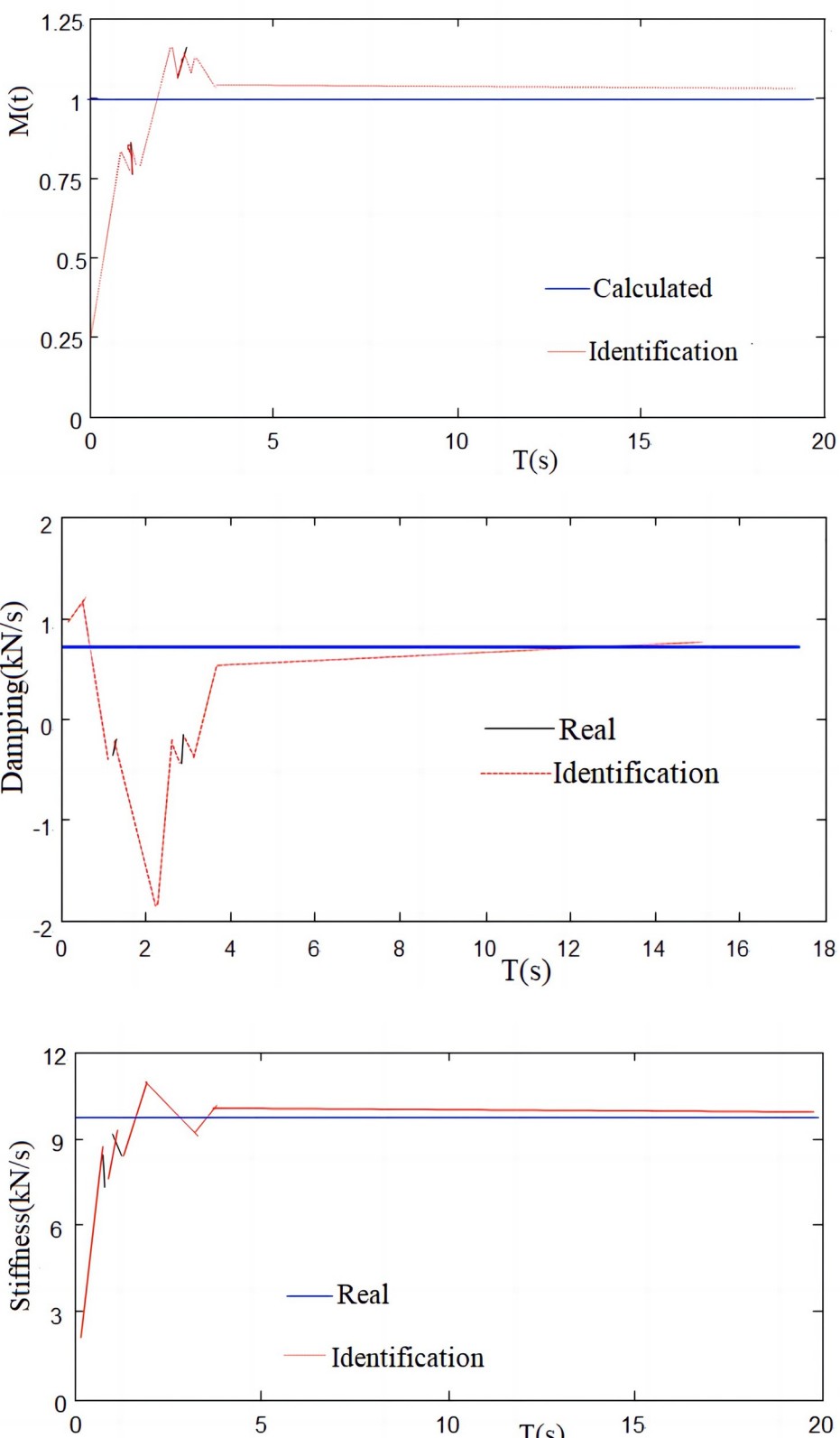

**Fig 10. Identification and comparison of simulation curves.** (a) Quality recognition and calculated simulation results; (b) Damp recognition and calculated simulation results; (c) Stiffness recognition and calculated simulation results.

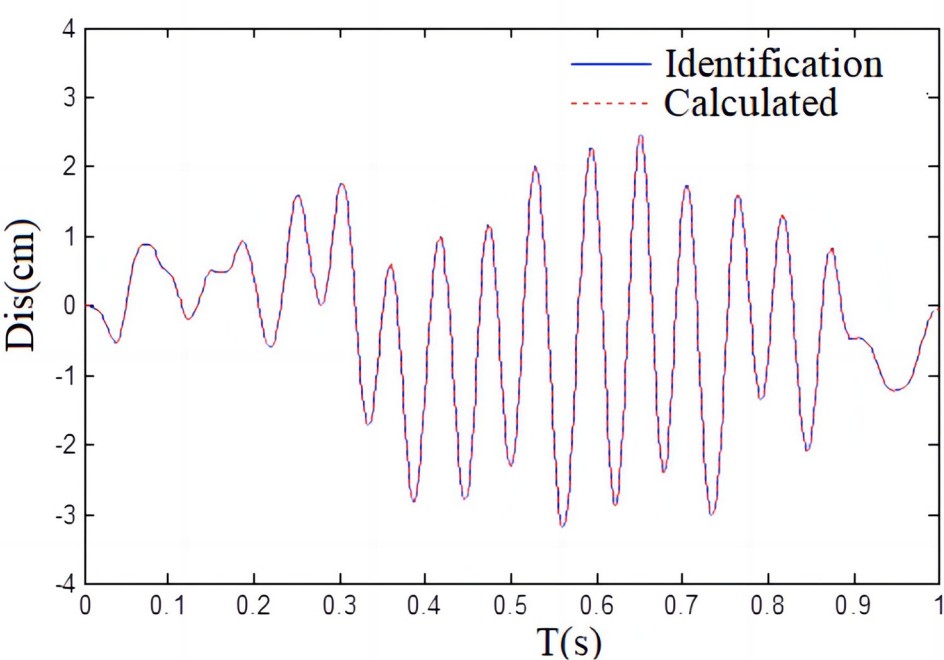

**Fig 11. Output displacement of identification compared with the calculated value.**

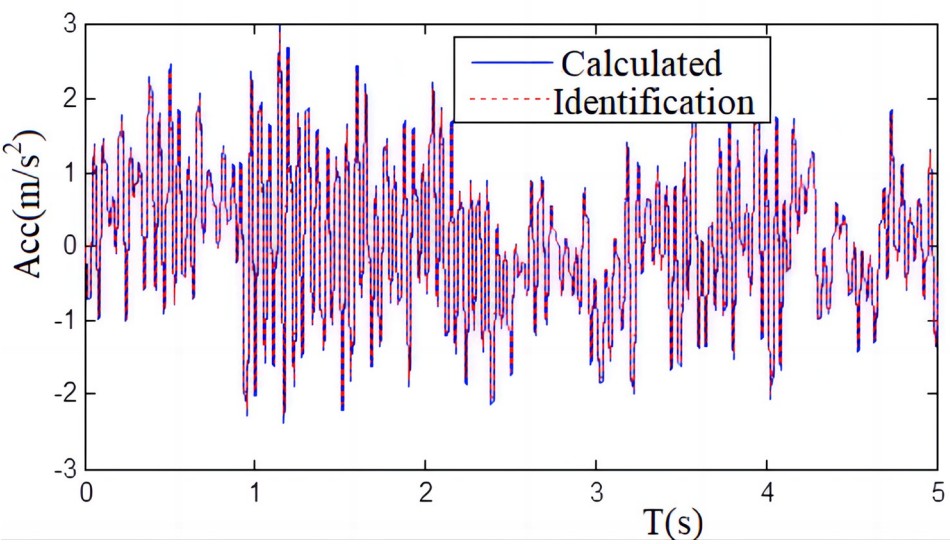

**Fig 12. Acceleration measurement results compared with the calculated results.**

**Table 5. Test condition table.**

| condition | Seismic wave | duration | peak displacement (cm) | Peak acceleration (g) |
|-----------|--------------|----------|------------------------|-----------------------|
| 1 | EL-Centro (NS) | 20s | 11 | 0.33 |
| 2 | EL-Centro (NS) | 20s | 2.5 | 0.22 |
| 3 | EL-Centro (NS) | 20s | 1.4 | 0.15 |

that the recognition method proposed in this paper can be applied to the seismic simulation shaking table in the parameter identification of the matrix system and be used to establish a proper model.

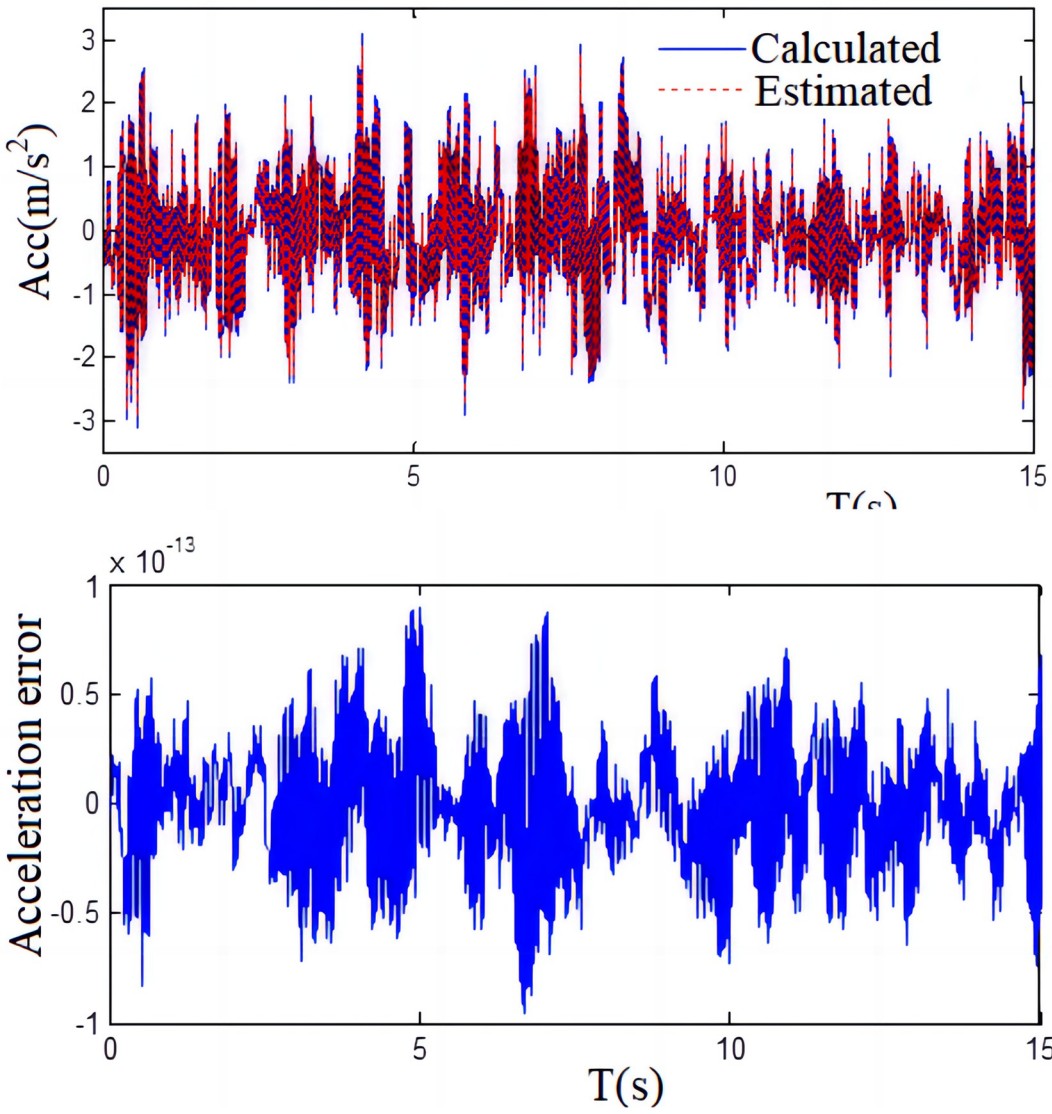

**Fig 13. Curve comparing the calculated value of acceleration with the measured value.** (a) Comparison of the calculated and measured responses; (b) Calculated acceleration and acceleration measurement error values.

**Table 6. Identification results of the shaking table array.**

| | Mass/t | stiffness/(kN/m) | damping/(kN/m/s) |
|---|---|---|---|
| 1 | 1.132 | 9.406 | 0.932 |
| 2 | 1.138 | 9.428 | 0.941 |
| 3 | 1.140 | 9.437 | 0.949 |

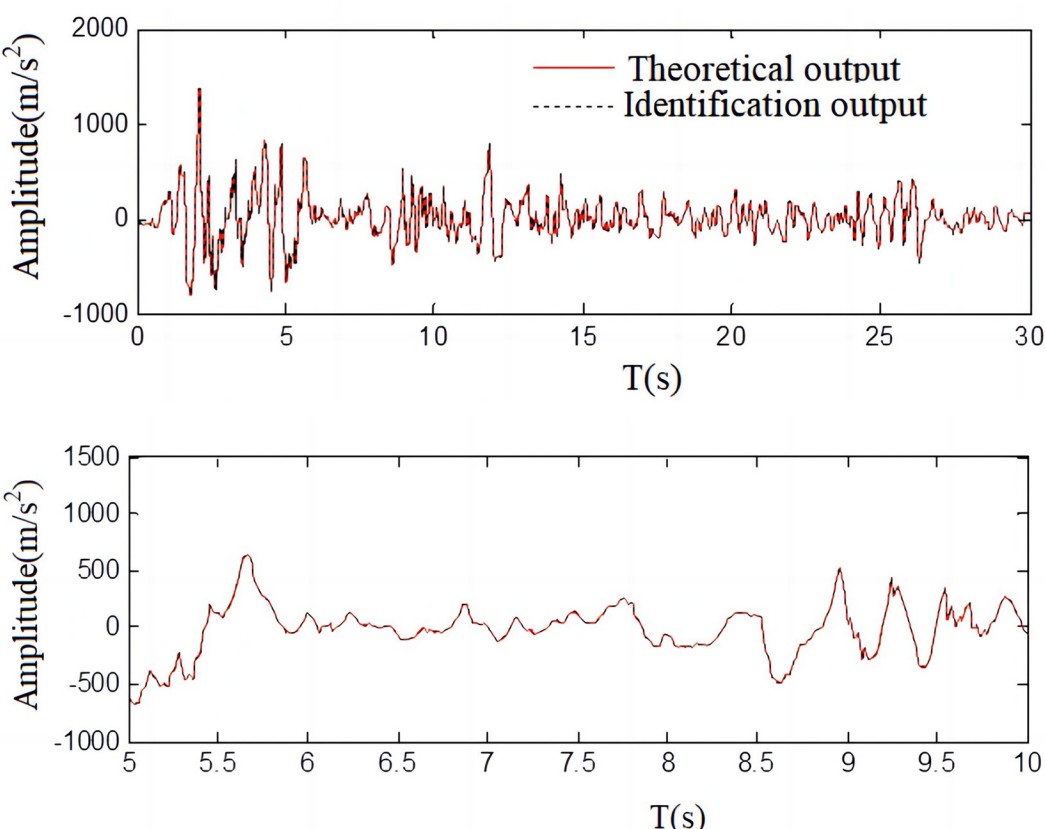

**Fig 14. Acceleration time history curve.** (a) Acceleration time history curve; (b) Local magnification of acceleration time-history curve.

## 5. Conclusion and discussion

The existence of nonlinear factors such as servo valve dead zone, friction and clearance between components leads to a certain degree of uncertainty in the shaking table system model, and the waveform tracking accuracy is not high. Recursive least square method is widely used in parameter identification because of its simplicity and high efficiency. In this paper, it is applied to the parameter identification of nine sub-stations.

In this paper, the model of a shaking table array system was built, and the parameters of mass, damping and stiffness were identified by the RLS identification algorithm. The identification model was verified by the shaking table array system test, and the comparison between the measured response and the calculated response of the landing surface with different loading modes verifies the correctness of the recursive least squares identification algorithm for the identification of array system parameters.

The research work in this paper is based on closed-loop system and single-degree-of-freedom structure. The subsequent research may consider verifying the applicability of RLS method to open-loop system and multi-degree-of-freedom structure, and further improve the efficiency and accuracy of system identification by combining with other intelligent control algorithms.

## Acknowledgments

The author thanks the teachers and classmates of the team for collecting the experimental data.

## Author Contributions

**Data curation:** Mengyuan Qin, Cun Li, Zihan Yuan.

**Software:** Cun Li.

**Validation:** Zihan Yuan.

**Writing – original draft:** Chunhua Gao.

**Writing – review & editing:** Chunhua Gao, Yanping Yang.

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
