## [Decision Letter · Decision Letter 0]

4 Nov 2022

PONE-D-22-26985Research on parameter identification of shaking table systems based on the RLS methodPLOS ONE

Dear Dr. yang,

Thank you for submitting your manuscript to PLOS ONE. After careful consideration, we feel that it has merit but does not fully meet PLOS ONE’s publication criteria as it currently stands. Therefore, we invite you to submit a revised version of the manuscript that addresses the points raised during the review process.

We look forward to receiving your revised manuscript.

Kind regards,

Muhammad Usman

Academic Editor

PLOS ONE

“The author would like to thank the Henan Provincial Department of Science and Technology for the scientific and technological research project (182102210539) and the Key Scientific Research Project of Henan Province (No 18B560009), the South Lake Young Scholars Program of Southwest Normal University, the Graduate Innovation Fund Project of Xinyang Normal University (Nos 2021KYJJ75), and the School of Xinyang Normal University Architecture and Civil Engineering Laboratory. The author thanks the teachers and classmates of the team for collecting the experimental data.”

“GCH was supported by the Key Science and Technology Project of the Science and Technology Department of Henan Province (182102210539), the Key Research Project of Henan Province (No 18B560009) and the Nanhu Young Scholars Program of Southwest Normal University, while YYP was supported by the Graduate Innovation Fund Project of Xinyang Normal University (No  2021KYJJ75).

Reviewers' comments:

Reviewer's Responses to Questions

**Comments to the Author**

1. Is the manuscript technically sound, and do the data support the conclusions?

Reviewer #1: Yes

Reviewer #2: Yes

Reviewer #3: Yes

2. Has the statistical analysis been performed appropriately and rigorously? 

Reviewer #1: Yes

Reviewer #2: Yes

Reviewer #3: Yes

3. Have the authors made all data underlying the findings in their manuscript fully available?

Reviewer #1: Yes

Reviewer #2: Yes

Reviewer #3: No

4. Is the manuscript presented in an intelligible fashion and written in standard English?

Reviewer #1: Yes

Reviewer #2: Yes

Reviewer #3: Yes

5. Review Comments to the Author

Reviewer #1: The manuscript presents some noteworthy outcomes regarding the use of a seismic simulator to test model resiliency to earthquake loads. However, few recommendations are made, nevertheless, to improve the quality even further.

1. Please clarify if such is applicable for an open-loop control system (non-feedback system), in which the control action is unaffected by the output OR for a closed-loop control system (feedback system), that modifies the state based on the output.

2. Does the transfer function remains the same even if the stiffness of the test model changes over time, for example, in case of incremental dynamic analysis. Please refer to some shaking table testing: https://doi.org/10.1080/13632469.2017.1326426

3. Please improve quality of the figures.

4. Please provide clear recommendations in the conclusions.

5. Also, state the scope and limitation of the work. How the system will be extended to system with MDOFs.

Reviewer #2: General Comments:

1. In abstract spelling of keywords need to be fix.

2. Number of keywords should be at least five.

3. In Figure-04, at few locations text is not readable so increase the font size.

Conclusions and Recommendations:

It is recommended to divide the conclusions into three paragraphs. In the first para, state about the problem of investigation and your methodology in the current study. In the second para, describe about results and in the last paragraph add some recommendations for future line of action.

Reviewer #3: Comments:

The authors identify the system key parameters of the shaker based on the recursive least squares algorithm and the paper has some research implications. However, there are some shortcomings in the authors' study, and the authors are recommended to revise the paper carefully. The specific comments are as follows.

1. The introduction is too lengthy on the background of the study and the review of similar studies cites too little literature; the authors are advised to revise the introduction carefully.

2. The specific structure of the research object is too little described in the second part, and the pictures are not clear. From the introduction of the paper, it is clear that the vibration system is in an array structure and the authors could have made a specific presentation from a single vibration module.

3. The recursive least squares identification algorithm is a classical identification algorithm, and it is not necessary to use a large space in the paper to introduce the computational process, and it is sufficient to add references.

4. The authors do not provide a specific analysis of why the RLS algorithm is used. Among the classical recognition algorithms, there are "support vector machines", "neural networks", "Bayesian estimation", and so on. What are the differences between these algorithms and RLS algorithms in the application scenario of this study, and what are the advantages and disadvantages of each of them, and what are the advantages and disadvantages of each of them by comparison? It is suggested that the authors add the comparison of recognition algorithms.

5. The article also has many minor problems, for example, the formula numbers should be uniformly right-justified, the typographical line spacing is too large, and some expressions are too verbose, so I suggest the authors carefully revise the whole article.

6. PLOS authors have the option to publish the peer review history of their article (what does this mean?). If published, this will include your full peer review and any attached files.

Reviewer #1: No

Reviewer #2: No

Reviewer #3: No

---

## [Author Response · Author response to Decision Letter 0]

19 Nov 2022

Article number: PONE-D-22-26985

Title：Research on parameter identification of shaking table systems based on the RLS method

First of all, I would like to thank the reviewer and editor for taking time out of their busy schedule to review my article and put forward some valuable suggestions. All authors would like to express their great gratitude!  All the questions and suggestions raised by reviewers and editors have been modified and improved in the submitted revision manuscript, and the specific modification content is explained as follows:  

According to the requirements of the journal, we have modified the manuscript format and deleted the content related to the funding information in the acknowledgments. We have changed the funding statement and the revised funding statement is as follows:

“GCH was supported by the Key Science and Technology Project of the Science and Technology Department of Henan Province (182102210539), the Key Research Project of Henan Province (No 18B560009), (No 23A560015) and the Nanhu Young Scholars Program of Southwest Normal University, while YYP was supported by the Graduate Innovation Fund Project of Xinyang Normal University (No 2021KYJJ75).

Reviewer 1 revision instructions:

1、Please clarify if such is applicable for an open-loop control system (non-feedback system), in which the control action is unaffected by the output OR for a closed-loop control system (feedback system), that modifies the state based on the output.

This study is applicable to the closed-loop control system, and the state is improved according to the system output each time, so that the output each time is closer to the actual engineering situation. The open-loop control system is not verified in this paper. A trial study can be conducted in the follow-up study.

2、Does the transfer function remains the same even if the stiffness of the test model changes over time, for example, in case of incremental dynamic analysis. Please refer to some shaking table testing: https://doi.org/10.1080/13632469.2017.1326426

According to the reviewer's opinion, the supplement was made in the manuscript.

Ahmad et al. [7] established a scale model of a two-story reinforced concrete frame structure with a 1/3 scale coefficient, and conducted a full dynamic shaking table test on it. The study found that when low-strength concrete was used and no node limit band was set, the safety of the joint was difficult to be guaranteed. It is verified that the ASCE ACI 352 node shear strength model overestimates the shear strength of low strength concrete.

3、Please improve quality of the figures.

According to the reviewer's suggestion, the figures in the manuscript were processed.

Fig. 1 Array system

Fig. 2 Mesa support system

Fig. 3 Single shaking table movement mode

Fig. 4 Typical arrangement of the array table

Fig. 5 The control system

Fig. 6 Schematic diagram identification system using the least squares method

Fig. 7 Curve relationship with normalized average error and different parameters

(a) (b)

Fig. 8 Coulomb friction curve under different working conditions

(a) (b)

Fig. 9 System damping coefficient curve under different conditions

(a)Quality recognition and calculated simulation results

(b)Damp recognition and calculated simulation results

(c)Damp recognition and calculated simulation results

Fig. 10 Identification and comparison of simulation curves

Fig. 11 Output displacement of identification compared with the calculated value

Fig. 12 Acceleration measurement results compared with the calculated results

(a) Comparison of the calculated and measured responses

(b) Calculated acceleration and acceleration measurement error values

Fig. 13 Curve comparing the calculated value of acceleration with the measured value

(a)Acceleration time history curve

(b) Local magnification of acceleration time-history curve

Fig. 14 Acceleration time history curve

4、Please provide clear recommendations in the conclusions.

According to the reviewer's opinion, the conclusion of the manuscript is supplemented.

The research work in this paper is based on closed-loop system and single-degree-of-freedom structure. The subsequent research may consider verifying the applicability of RLS method to open-loop system and multi-degree-of-freedom structure, and further improve the efficiency and accuracy of system identification by combining with other intelligent control algorithms.

5、Also, state the scope and limitation of the work. How the system will be extended to system with MDOFs.

In this paper, the RLS method is used to identify the parameters of seismic simulation shaking table. This identification method is applicable to both single and multi-degree-of-freedom systems, but the author's research has not been extended to multi-degree-of-freedom systems. In the subsequent research work, the author and his research group can explore and study the multi-degree-of-freedom systems.

Reviewer 2 revision instructions:

1．In abstract spelling of keywords need to be fix.

According to the reviewer's comments, the key words of the manuscript were revised as follows: shaking table, nine sub-table system, system identification, RLS method, single degree of freedom structure

2．Number of keywords should be at least five.

According to the reviewer's comments, the number of keywords in the manuscript was increased.

3．In Figure-04, at few locations text is not readable so increase the font size.

According to the reviewer's comments, Fig. 4 was modified, as shown in Fig.5 after modification.

Fig. 5 The control system

Conclusions and Recommendations:

It is recommended to divide the conclusions into three paragraphs. In the first para, state about the problem of investigation and your methodology in the current study. In the second para, describe about results and in the last paragraph add some recommendations for future line of action.

The conclusions in the paper have been revised according to the reviewer's suggestion; (black is the original content, red is the revised content)

Conclusion and discussion

In this paper, the model of a shaking table array system was built, and the parameters of mass, damping and stiffness were identified by the recursive least squares identification algorithm. The identification model was verified by the shaking table array system test, and the comparison between the measured response and the calculated response of the landing surface with different loading modes verifies the correctness of the recursive least squares identification algorithm for the identification of array system parameters. This study provides an important reference for establishing an accurate shaking table array system.

Conclusion and discussion

The existence of nonlinear factors such as servo valve dead zone, friction and clearance between components leads to a certain degree of uncertainty in the shaking table system model, and the waveform tracking accuracy is not high. Recursive least square method is widely used in parameter identification because of its simplicity and high efficiency. In this paper, it is applied to the parameter identification of nine sub-stations.

In this paper, the model of a shaking table array system was built, and the parameters of mass, damping and stiffness were identified by the RLS identification algorithm. The identification model was verified by the shaking table array system test, and the comparison between the measured response and the calculated response of the landing surface with different loading modes verifies the correctness of the recursive least squares identification algorithm for the identification of array system parameters.

The research work in this paper is based on closed-loop system and single-degree-of-freedom structure. The subsequent research may consider verifying the applicability of RLS method to open-loop system and multi-degree-of-freedom structure, and further improve the efficiency and accuracy of system identification by combining with other intelligent control algorithms.

Reviewer 3 revision instructions:

1.The introduction is too lengthy on the background of the study and the review of similar studies cites too little literature; the authors are advised to revise the introduction carefully.

According to the reviewer's opinion, the introduction was revised.

1.Introduction

The shaking table is a very important device for simulating earthquakes in the laboratory and provides an accurate method for measuring structural responses under seismic conditions [1-4]. With the development of the shaking table, it has become an effective measure for studying the elastoplastic seismic response of structures[5].According to different driving sources, it can be divided into electromagnetic drive and actuator drive, among which the actuator drive vibration table includes electric vibration table and electro-hydraulic servo vibration table[6]. Electro-hydraulic servo shaking table is widely used in various fields, such as civil, aerospace and nuclear engineering, because of its advantages of large output, large stroke and fast response.Ahmad et al. [7] established a scale model of a two-story reinforced concrete frame structure with a 1/3 scale coefficient, and conducted a full dynamic shaking table test on it. The study found that when low-strength concrete was used and no node limit band was set, the safety of the joint was difficult to be guaranteed. It is verified that the ASCE ACI 352 node shear strength model overestimates the shear strength of low strength concrete.

However, in the modelling process of the electrohydraulic servo system of the hydraulic vibration table, there are many nonlinear factors that cannot be well reflected in the model, such as the dead zone of the servo valve, friction between components, and clearance, which cause the system model to exhibit a considerable degree of uncertainty[8]. In addition, the dynamic characteristics of the servo system are also affected by factors such as equipment manufacturing, oil supply pressure, and level of input signal, so that the output waveform of the exciter is different from the expected waveform, and the waveform reproduction ability is weak[9-10]. To solve these problems, many scholars have proposed an identification method to determine the parameters that are difficult to calculate.

Many identification methods are commonly used in engineering, such as neural network, support vector machine, Bayesian estimation and least square method[11-13]. Among them, the neural network algorithm adjusts the similarity between the output and the expectation by adjusting the connection weight between neurons, which requires a large number of data sets, and its own interpretability is relatively poor, and it is easy to fall into the local optimal solution [14-15]. Based on statistical theory and structural risk minimization principle, support vector machine algorithm has global optimality and good "robustness". However, it is difficult to implement large-scale training samples, and is sensitive to the selection of parameters and kernel functions, with great uncertainty [12,16]. Bayesian estimation uses Bayesian theorem combined with prior information and new observation information to estimate the new probability of occurrence, which to some extent solves the problem that classical estimation cannot be applied to solving the probability of non-repeatable independent events. However, it relies on the distribution state of prior information and has poor objectivity [17]. The least squares method is widely used in the field of system model parameter identification by minimizing the sum of squared errors to obtain the best matching function, which is simple, good analytical and unique in optimal solution[13,18-19].

Compared with the least square method, the recursive least square method (RLS) can be used for real-time identification with less computational effort. Luo qinqin et al[20]. identified the main parameters of the virtual synchronous generator by using recursive least square method, and the results show that the identified parameters meet the accuracy requirements. Lin Juguang et al[21]. proposed an improved recursive least square method for online identification of the inductance parameters of the D-axis and the Q-axis of permanent magnet synchronous motor, and achieved efficient identification results. In this paper, nine sub-table system is taken as the research object, and RLS algorithm is used to identify inherent parameters of the shaking table that will change during the experiment, such as the mass and stiffness. Through different test data, the parameters of shaking table are estimated by RLS algorithm, and the response of shaking table is rebuilt. The surface response of shaking table reconstructed by RLS method is compared with the surface response of shaking table measured by test, and the final model of shaking table is determined. It is found that the identification of the parameters of the shaking table by the RLS method is accurate and available and can provide an important reference for future shaking table research.

2.The specific structure of the research object is too little described in the second part, and the pictures are not clear. From the introduction of the paper, it is clear that the vibration system is in an array structure and the authors could have made a specific presentation from a single vibration module.

According to the reviewer's comments, the relevant content has been supplemented in the second part of the manuscript and the definition of the picture has been revised.

This paper takes the nine-subarray shaking table system of the Beijing University of Technology as the research object, as shown in Fig. 1. The table support system is shown in Fig. 2. The modular array system can be placed freely to form a large vibration table for testing or can be used for multiple shaking tables for array testing. Each sub-shaker consists of four components, namely the table, shaker, connecting rod and base, and each sub-shaker can be mounted vertically or horizontally at any time to achieve different seismic wave input and motion modes. For example, a shaker is installed in the x direction, two connecting rods in the y direction, and three connecting rods in the z direction to achieve unidirectional motion, as shown in Fig. 3 (a); Install one shaker and two shakers in the x direction and y direction respectively, and install three connecting rods in the z direction to realize the three degrees of freedom movement of double horizontal plus rotation, as shown in Fig. 3 (b); By installing one shaker in the x direction, two shakers in the y direction and three shakers in the z direction, three-direction and six-DOF motion can be realized, as shown in Fig. 3 (c)[22]. The layout of array system is flexible, and the typical layout of the array is shown in Fig. 4. Table 1 shows the skill performance indices of the nine subarray systems. Fig. 5 is a conceptual diagram of the entire control system, which is a host-target style control system. The host computer serves as a human-computer interaction interface and allows users to enter command signals and displays the system’s current state and data.

Fig. 1 Array system

Fig. 2 Mesa support system

Fig. 3 Single shaking table movement mode

Fig. 4 Typical arrangement of the array table

Fig. 5 The control system

3.The recursive least squares identification algorithm is a classical identification algorithm, and it is not necessary to use a large space in the paper to introduce the computational process, and it is sufficient to add references.

The manuscript has been revised according to the reviewer's comments.

4．The authors do not provide a specific analysis of why the RLS algorithm is used. Among the classical recognition algorithms, there are "support vector machines", "neural networks", "Bayesian estimation", and so on. What are the differences between these algorithms and RLS algorithms in the application scenario of this study, and what are the advantages and disadvantages of each of them, and what are the advantages and disadvantages of each of them by comparison? It is suggested that the authors add the comparison of recognition algorithms.

According to the reviewer's opinion, a brief introduction and analysis of advantages and disadvantages of "support vector machine", "neural network", "Bayesian estimation" and "least square method" are added in the introduction.

Many identification methods are commonly used in engineering, such as neural network, support vector machine, Bayesian estimation and least square method[10-12]. Among them, the neural network algorithm adjusts the similarity between the output and the expectation by adjusting the connection weight between neurons, which requires a large number of data sets, and its own interpretability is relatively poor, and it is easy to fall into the local optimal solution [13-14]. Based on statistical theory and structural risk minimization principle, support vector machine algorithm has global optimality and good "robustness". However, it is difficult to implement large-scale training samples, and is sensitive to the selection of parameters and kernel functions, with great uncertainty [11,15]. Bayesian estimation uses Bayesian theorem combined with prior information and new observation information to estimate the new probability of occurrence, which to some extent solves the problem that classical estimation cannot be applied to solving the probability of non-repeatable independent events. However, it relies on the distribution state of prior information and has poor objectivity [16]. The least squares method is widely used in the field of system model parameter identification by minimizing the sum of squared errors to obtain the best matching function, which is simple, good analytical and unique in optimal solution[12,17-18].

5．The article also has many minor problems, for example, the formula numbers should be uniformly right-justified, the typographical line spacing is too large, and some expressions are too verbose, so I suggest the authors carefully revise the whole article.

According to the reviewer's comments, all formulas, charts and typesetting of the full text have been comprehensively checked and modified; Double line spacing is chosen according to the submission guide.

According to the suggestions of reviewers and editors, the whole paper has been revised, and make the above explanation to the revision of the manuscript,please experts and editors review! And if you find any further deficiencies during the review, Please let us know. if the revision meets the requirements, the paper is expected to be published as soon as possible.

Once again to express heartfelt thanks for the reviewer's and editor's help!

Author：Chunhua Gao, Yanping Yang, Mengyuan Qin, Cun Li, Zihan Yuan

College of Architecture and Civil Engineering 

Xinyang Normal University

Xinyang

China

---

## [Decision Letter · Decision Letter 1]

1 Dec 2022

Research on parameter identification of shaking table systems based on the RLS method

PONE-D-22-26985R1

Dear Dr. yang,

We’re pleased to inform you that your manuscript has been judged scientifically suitable for publication and will be formally accepted for publication once it meets all outstanding technical requirements.

Kind regards,

Muhammad Usman

Academic Editor

PLOS ONE

Additional Editor Comments (optional):

Reviewers' comments:

Reviewer's Responses to Questions

**Comments to the Author**

1. If the authors have adequately addressed your comments raised in a previous round of review and you feel that this manuscript is now acceptable for publication, you may indicate that here to bypass the “Comments to the Author” section, enter your conflict of interest statement in the “Confidential to Editor” section, and submit your "Accept" recommendation.

Reviewer #3: All comments have been addressed

2. Is the manuscript technically sound, and do the data support the conclusions?

Reviewer #3: Yes

3. Has the statistical analysis been performed appropriately and rigorously? 

Reviewer #3: Yes

4. Have the authors made all data underlying the findings in their manuscript fully available?

Reviewer #3: Yes

5. Is the manuscript presented in an intelligible fashion and written in standard English?

Reviewer #3: Yes

6. Review Comments to the Author

Reviewer #3: I think that the paper can be published now.

The authors addressed all my comments.

The paper is recommended for publication.

7. PLOS authors have the option to publish the peer review history of their article (what does this mean?). If published, this will include your full peer review and any attached files.

Reviewer #3: No

---

## [Editor Report · Acceptance letter]

15 Dec 2022

PONE-D-22-26985R1 

Research on parameter identification of shaking table systems based on the RLS method 

Dear Dr. yang:

I'm pleased to inform you that your manuscript has been deemed suitable for publication in PLOS ONE. Congratulations! Your manuscript is now with our production department. 

Kind regards, 

on behalf of

Dr. Muhammad Usman 

Academic Editor

PLOS ONE